# Key factors influencing the prescribing of statins: a qualitative study among physicians working in primary healthcare facilities in Indonesia

Sylvi Irawati [1,2,3] Sari Prayudeni,[4] Riris Rachmawati,[5] I Wayan Wita,[6] Bob Willfert,[1,7] Eelko Hak,[1] Katja Taxis[1]

SI and SP are joint shared authors.

## ABSTRACT

**Objectives** To elicit key factors influencing physicians' decision to prescribe statins.

**Design** A qualitative study using a phenomenological approach within a pragmatism interpretive framework. A combination of purposive and snowball sampling was used to recruit physicians. Data were collected through face-to-face, semistructured interviews with physicians working in primary healthcare facilities in a capital of a province in Indonesia. We recorded and verbatim transcribed the interviews. Coding was done independently by two researchers and data were analysed using phenomenological data analyses. Key factors influencing physicians' decision to prescribe statins were classified into factors at the microlevels, mesolevels and macrolevels according to the structural model by Scoggins *et al.*

**Participants and setting** Physicians working in primary healthcare facilities in a capital of a province in Indonesia.

**Results** Ten physicians were included in the study. Key factors at the microlevel were that physicians knew guidelines in general, but there was uncertainty how to take into account the level of total cholesterol in combination with other cardiovascular risk factors such as diabetes and hypertension. At the macrolevel, the new National Health Insurance System (NHIS) that appeared to facilitate the prescription of statins though more clinical information should be integrated in the system's platform to support appropriate prescribing.

**Conclusions** The findings indicate lack of awareness of specific details in current guideline recommendations. Appropriate prescribing of statins should be enhanced using the new NHIS.

## Strengths and limitations of this study

► This is the first study to gain insights from physicians on key factors influencing their decision in prescribing statins in a lower middle-income country setting.
► Face-to-face interviews by an interviewer who was known to physicians made it possible to gain more trustworthy information.
► Follow-up interviews, member checking process, independent coders, discussion with several researchers to reach consensus and the use of a theoretical framework improved the credibility of the results.
► Despite code saturation was appropriately reached with a small sample size, a larger sample size is needed to obtain meaning saturation to provide a deeper understanding of the investigated phenomenon.

## INTRODUCTION

The use of statins has been endorsed by local and various international guidelines[1–6] to prevent cardiovascular (CV) disease (CVD)-related events or death in individuals without history of previous CVD (primary prevention) or with established CVD (secondary prevention).[7–9] But many studies demonstrate underprescribing, and in some cases overprescribing of this drug class. Underprescribing of statins is especially an issue in primary prevention,

for example, in the groups with moderate-to-very high risk of CVD in the European general population and in an outpatient setting in the USA.[10–13] One small study (n=243 cases) with medical records data taken from three private hospitals in Jakarta, the capital city of the Republic of Indonesia, a lower middle-income country (LMIC), suggested patients might not be properly treated with statins. But underprescribing or overprescribing was not clearly defined, because data required to determine the appropriateness to prescribe statins are not routinely measured or well documented.[14 15] Another study in ambulatory patients with diabetes with established CVD in Surabaya revealed the phenomenon of underprescribing with 43% of eligible patients not receiving statins.[16] Studies investigating the consequences of underprescribing and overprescribing of statins are scarce. Nevertheless, one case study in a UK population with a ≥20% of 10-year risk of developing CVD, indicating the use of statins for primary prevention of

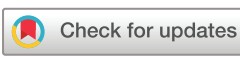

For numbered affiliations see end of article.

**Correspondence to**
Sylvi Irawati; s.irawati@rug.nl

CVD, illustrates that underprescribing will substantially reduce quality-adjusted life years gained over a 9-year period (2006–2014) and overprescribing will certainly increase the cost burden of prescription between £80 million and £117 million annually without certainty whether the benefit of prevented CVD events outweighs the adverse drug reaction.[17]

Factors influencing physicians' prescribing behaviour, regardless of the medication being prescribed, have been investigated since 1950s. Physicians' personal and professional characteristics, pharmaceutical marketing strategies (direct-to-physician or direct-to-consumer marketing), drug information sources, drug characteristics, drug price and patients' characteristics have been identified as factors influencing physicians' prescribing behaviour, although there is inconsistency over the relative importance of each factor.[18–24] More importantly, the majority of studies in this field have been conducted in high-income Western countries with a different culture, healthcare and health insurance systems compared with LMICs.

In order to reach universal health coverage (UHC) in 2019, the Government of Indonesia launched a new mandatory National Health Insurance System (NHIS) programme (*Jaminan Kesehatan Nasional-Kartu Indonesia Sehat* (*JKN-KIS*)) run by the Indonesian Social Security Agency for Health (*Badan Pengelola Jaminan Sosial, BPJS Kesehatan*) in 2014.[25–27] The system facilitates more patients to go to the primary healthcare facilities first instead of directly visiting specialists in the secondary facilities.[25 28] Moreover, more medicines, listed in the 2014 revised edition of the National Formulary, are insured by the government to improve access to medications for patients.[25 29 30] Based on the National Formulary, two types of statins are insured (pravastatin and simvastatin). An estimation of the total CV risk score, which is necessary to guide statin prescribing for primary prevention of CVD, is not mentioned.[29 30] This policy is different compared with the new clinical guideline for patients with dyslipidaemia issued by the Indonesian Heart Association (IHA) which adapted guidelines issued by the European Society of Cardiology and the European Atherosclerosis Society.[2 31] In these guidelines, statins should be prescribed according to both the level of low-density lipoprotein cholesterol (LDL-c) and the total CV risk score, especially in patients with unknown or no history of CVD. Thus, this situation may create confusion for physicians in prescribing statins and increase the risk of underprescribing or overprescribing of statins. In the current qualitative study, we aimed to understand the key factors influencing physicians to prescribe statins, especially in the primary healthcare facilities in an LMIC.

## METHODS

### Study design and framework

This was a qualitative study using a phenomenological approach within a pragmatism interpretive framework. The pragmatism framework focuses on the phenomenon being investigated (key factors influencing physicians in prescribing statins) and the question inquired regarding this phenomenon. This framework allowed choosing the method that worked best in the field to answer the research question.[32] We used the Standards for Reporting Qualitative Research checklist when writing our report.[33]

### Study context and setting

As briefly introduced earlier, the Government of Indonesia set a new NHIS conducted by *BPJS Kesehatan*, namely *JKN-KIS*, in 2014. In this system, Indonesia's Ministry of Health also enforced the already existing platforms such as the National Formulary and the e-Catalogue to support cost-effective medicine and medical supply procurement throughout the country. The National Formulary lists medicines reimbursed by BPJS. The e-Catalogue is a national electronic platform required to be used by healthcare facilities for procuring medicines. The e-Catalogue lists medicines with their prices and selected pharmaceutical companies as supplier. Unless the medicines which are needed are not listed in the e-Catalogue, it is mandatory for all healthcare facilities to purchase medicines from this platform.[25 26]

Indonesia's primary healthcare is delivered by public and private primary healthcare facilities. Public primary healthcare facilities consist of primary health centres (PHCs or *puskesmas*) and their supporting networks (*pustu, puskel, polindes, poskesdes*) owned and organised by District or City Health Offices. Private primary healthcare facilities consist of individual practice general practitioners (GPs), midwives, and nurses; and private clinics. As a part of new NHIS program in 2014, the national referral system to access these facilities was also established. Figure 1 presents an overall picture of the healthcare delivery system in Indonesia.[34]

### Participants

We viewed the topic of our study was sensitive in nature.[35 36] Prescribing activity in Indonesia is an authority of registered physicians governed by regulations[37 38] and ethical codes.[39 40] There might be issues which would not be frankly expressed due to its sensitivity. Thus, we selected a city, a capital of a province in Indonesia, where our study had



**Figure 1** Healthcare delivery system in Indonesia, figure was modified from Claramita *et al*.[34] *Puskesmas* is a public primary health centre. *Posyandu* is a service organised by volunteers (*Kader*) to deliver health programme for vulnerable groups, especially mother and children, locally. *Posyandu* is supervised by *Puskesmas*.[52]

been welcomed and approved by the local physician association of the city. We gathered information from physicians working at primary care health facilities in this city. The province was quite small with an area around 6000 ha. The city is the most populous city in the province with around 900 000 inhabitants in 2017, about 21.5% of the province's population. Indonesian regulation allows physicians to conduct their practice at a maximum of three healthcare facilities (public or private, primary or secondary level).[38] The views of the physicians included in our study reflect their practice to care for patients with and without CVD with possible indications for primary and secondary prevention of CVD.

We obtained the list of physicians' names registered as member of the physician association of the city, but up-to-date physicians' personal contact details were not available. Therefore, we purposively selected physicians who worked at the primary healthcare facilities in the capital city who were known to a member of the research team (SP) in a personal (friendship) or professional (acquaintance) manner, and consented to participate in the study. Participating physicians were then asked for contact details of potential other participants (snowball sampling).

### Data collection

Baseline characteristics of the participants were collected by interview (age, practice experience, educational period). Initially, an interview guide was developed by SI and SP based on general key factors in prescribing found in the literature (see box 1).[18] A semistructured face-to-face interview was conducted with each participant by SP between July 2015 and January 2016. To obtain the key factors considered by physicians in prescribing statins, SP only asked the main question at first. The probing questions were asked

---

**Box 1    Interview guide**

**Questions**

*Main question*

► Which factors do you consider when prescribing statins to the patient ?

*Probing question* (not necessarily being asked, depends on answers from the main question)

► Which therapeutic guideline did you usually use in prescribing statins ?

► How do you assess a total cardiovascular risk factor ?

► Which type of statins do you often prescribe ?

► Which statin dose and duration of use you prescribe ?

► How about the relation to pharmaceutical company promotion/advertising ?

► Does the regulation in the practice setting limit the decision of therapy ?

► Have you ever found patients complaining about the cost of statins ?

► Are you having difficulties in arranging time for a consultation with the patient because there are many queues ?

► Does an insurance system such as the new national health insurance programme limit you to prescribe statins to patients who need it ?

---

after physicians answered the first question in an explanatory way. All interviews were audio recorded using a tape recorder. The interviewer also took some field notes. The result from the initial interview of each participant was preanalysed directly by SI and SP to confirm whether it had covered the aim of the study and to determine if a follow-up interview with the same participant was needed. Data collection was discontinued when code saturation was reached as described by Hennink *et al*.[41]

### Data analysis

The recorded interviews were listened to repeatedly and transcribed verbatim whenever possible. The field notes were checked to add information to the transcripts. To enhance trustworthiness and credibility of data analysis, a technique known as member checking was applied by returning each transcript to the respective participant to confirm the content of the interview.

Using the phenomenological approach, we highlighted and coded keywords which indicated key factors in prescribing statins from the transcripts. The same code was assigned to the same keyword. We categorised the codes in minor subthemes, subthemes and final major themes. The encoding process until obtaining subthemes was conducted independently by SP and RR. Disagreement regarding the codes and subthemes was discussed between SI and SP to reach a consensus. After subthemes were drawn, SP, SI and IWW discussed major themes and compared the themes with the findings from similar qualitative studies.

The final themes were refined by SI and KT and were mapped into a model that had previously been developed by Scoggins *et al*.[42] This was a qualitative study, investigating factors influencing GPs to prescribe statins in England using in-depth interviews and focus group discussions. Interviews were done with senior managers primary care service organisations (primary care trusts); focus group discussions were done with GPs of those organisations. The factors were classified into macrolevel, mesolevel and microlevel (figure 2).

### Patient and public involvement

There was no patient and public involvement in the design, conduct or reporting of our research.

### RESULTS

Ten physicians (nine females, one male) were willing to participate in this study. The mean age of all physicians was 31.7 years old (ranged 24–56). Five physicians graduated between 2010 and 2015, four between 2000 and 2010, and one between 1980 and 1990. The mean duration of medical education was 6.6 years (ranged 5–12 years). All physicians were GPs. One was in the process of becoming a specialist in internal medicine. All physicians were working at more than one healthcare facility, four of them practised at three different healthcare facilities including at the secondary level. On average, the physicians examined around 99 patients per week (range 30–250); the mean number of patients with CVD risk was 26.0 patients weekly (range 5–100).

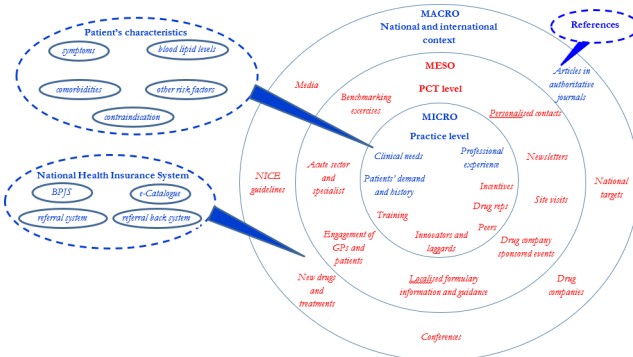

**Figure 2** Factors influencing general practitioners' (GPs) prescribing. Factors found from our study that matched to factors found by Scoggins et al[42] are presented in blue colour. BPJS, Badan Pengelola Jaminan Sosial; NICE, National Institute of Health and Clinical Excellence; PCT, primary care trust; reps, representatives.

Table 1 presents all major themes, subthemes and minor subthemes derived from transcript analysis. In general, there were two major themes drawn from the analysis: factors operating at (1) microlevel and (2) macrolevel. We did not identify any themes on the mesolevel. Figure 2 shows the themes mapped into the model by Scoggins et al.[42]

### Theme 1: key factors in statin prescribing at the microlevel

The physicians described the key factor influencing their decision to prescribe statins as their inherent understanding of the characteristics of patients which indicate the patient's need for a statin or a contraindication. There was some uncertainty around these characteristics which was inseparable from other factors like motivation and education. See for further explanation below.

### Subtheme: patient's characteristics

Physicians described several patient's characteristics which they considered to be important for prescribing statins. The most important characteristic was a high level of total cholesterol (TC); other characteristics were clinical symptoms, comorbidities and other risk factors.

From the level of cholesterol of the patient him/herself, if for example, it is more than 200 [mg/dL], for example 210, but it looks like it still can be [controlled] with dietary pattern, I have not prescribed it yet. However, if, for example, there have been symptoms such as [on the] neck and maybe he/she has high blood pressure, usually a medication [is prescribed]—simvastatin 1×10 mg. Actually, from my personal perspective, I do not look at the high blood pressure only. Only from the level of cholesterol, if, for example, it has been already high such as up to 250, 260, I give [the medication] even though the blood pressure is not high. (Physician 1)

Interestingly, there is disagreement over the cut-off of the TC level to initiate statins. Other disagreements were also found on how these factors related to each

| Major themes | Subthemes | Minor subthemes | Specific keywords |
|---|---|---|---|
| Microlevel | Patient's characteristics | Blood lipid levels | Total cholesterol (varied among participants: from 200 to 260 mg/dL), triglycerides, high LDL-c, low HDL-c, a combination of different lipid parameter |
| | | Symptoms | Pain (varied in location: back of the neck, neck, shoulder), stiffness (neck, shoulder), warmth (neck), tingling (palms, feet), dizziness |
| | | Comorbidities | Hypertension, diabetes, metabolic syndrome, heart disease |
| | | Contraindication | Pregnancy, liver disease |
| | | Other risk factors | Other risk factor without specifying the factor |
| | Professional experience | Primary and secondary prevention of CVD | Prevention, CVD prevention |
| Macrolevel | References | Standard | Standard |
| | | Guideline | Guideline from ACC/AHA |
| | | Journal | Article in journal |
| | National Health Insurance System | BPJS | BPJS |
| | | e-Catalogue | e-Catalogue |
| | | Referral system, referral back system | Patient referral |

**Table 1** Themes, subthemes and codings of factors influencing statin prescribing

ACC, American College of Cardiology; AHA, American Heart Association; BPJS, Badan Pengelola Jaminan Sosial; CVD, cardiovascular disease; HDL-c, high-density lipoprotein cholesterol; LDL-c, low-density lipoprotein cholesterol.

other when influencing the decision to prescribe statins. These variations of considerations were captured in the following quotes:

> It depends on the patient whether there is a comorbidity or not. If there is a comorbidity, usually [with the level of cholesterol] above 200, [statin] has already been prescribed. If not, maybe above 240. If there is hypertension, diabetes, above 200 also [it will] be directly given. (Physician 8)

> If, for example, the tryglycerides level is above 150, it can be prescribed. Then, after that… tryglycerides and total cholesterol. [For total cholesterol] usually above 200, it is prescribed. (Physician 10)

In prescribing statins, a physician also considered a situation where statin administration was contraindicated as quoted below:

> First, there has to be a clear or an appropriate indication. Then, whether there is a contraindication for the patient to use the medication for example in patients with liver disorder, then in patients who are pregnant—this is also a relative contraindication for statin administration. (Physician 6)

### Subtheme: professional experience of the prescribing physicians

This subtheme focused more on the personal choice of physicians. A mindset of CV event prevention motivated physicians to prescribe statins as described below:

> I give statins to the patient with high cholesterol lipid not just to decrease the level of the cholesterol. I mean that, you know that patient with high cholesterol lipid has much more risk for cardiovascular incident, because, in my knowledge, if the cholesterol level in the blood is so high, [it] can cause the plaque in the vascular, in the blood vessels. If the plaque becomes unstable, [it] can cause a rupture and make a mismatched perfusion and demand. So, I give the statin for the cholesterolemias, uh… the high cholesterol level in the laboratory test, other reason—no other reason, I think. It's the main reason. It's my first reason to give [is] to prevent. It prevents the cardio, the risk of cardiovascular incident. (Physician 4)

> … for statin administration as a medication to manage dyslipidemia, especially in patients with dyslipidemia or in patients with high risk of cardiovascular disease events that it can be as preventive therapy in cardiovascular disease, such as in patients who have already had a coronary heart disease or in patients with diabetes mellitus and… and with a risk, with peripheral artery disease so to say, hence there is a place to give [statin] or in patients with stroke. (Physician 6)

### Theme 2: key factors in statin prescribing operating at the macrolevel

We found that international information or references framed the physician's mindset of patients who were candidates to receive statins. Furthermore, the NHIS supported or hindered the decision to prescribe statins.

### Subtheme: references

References including journals and guidelines were information sources which may frame physicians' knowledge on which patients to prescribe statins for. These factors were not mentioned initially by participants but elicited through probing questions. The sources mentioned included therapeutic guidelines covered during medical education as well as local and global standards.

> I do… as in the standard operating procedure, that is, giving a healthy lifestyle first… a healthy dietary lifestyle. The standard operating procedure is from sources for example from journals… yes…also with medical references in the past… got from the college and when I was in the hospital [during internship as a junior physician] (Physician 2)

> In the past Framingham score was used to determine the risk for cardiovascular events. Now there is a new published guideline where [the risk] was not calculated from the Framingham score but [also] saw other risk factors. If not mistaken, [the guideline] was issued by American Heart Association also. (Physician 6)

> There is a standard for that. If, for example, [the level of cholesterol] is above 250, the medication is just started. If it is between 200–250, counseling related to dietary pattern and lifestyle only. (Physician 9)

### Subtheme: NHIS

Although the new NHIS means more patients get free access to medicines, there are still criteria to prescribe statins for reimbursement in the National Formulary which may or may not facilitate prescribing. For example, in the National Formulary, it was stated that pravastatin was to be given only to patients with hyperlipidaemia who had LDL-c level >160 mg/dL, to patients with coronary heart disease, and patients with diabetes and macroalbuminuria.[29 30] This suggested that physicians would tend to prescribe to this type of patients or would prescribe it anyway if the patient did not meet this strict criteria but was considered to need a statin with the consequence it would not be reimbursed by BPJS.

> The price is cheap, [it] is easy to obtain… in terms of it is available everywhere, then also available in generic. This also does not make it difficult for some patients even more when retired elderly patients do not have money but using *BPJS*. So, I think that is my consideration. (Physician 5)

> BPJS will only reimburse the administration of statins for LDL above 160. Therefore, if it is under 160, patients usually buy it by themselves, so we prescribe. Meanwhile, specialists in hospitals usually prescribe but this is not reimbursed by BPJS… so we help by prescribe it again. (Physician 8)

No, if indeed only the cholesterol is high, I think no need to refer [the patient]. No diabetes, nothing… we usually prescribe directly, unless when simvastatin is incidentally not available in *Puskesmas*… he/she [the patient] insists because he/she has a right… sometimes the medication is not available. He/she has the right… insurance right, for example he/she wants a referral, we refer. However, there he/she also does not certainly obtain [a statin] if the LDL is under 160. (Physician 8)

### A theme categorised under Pandora box

We attempted to elicit the impact of pharmaceutical companies and their marketing strategy related to statin prescribing, but physicians preferred to not discuss this topic with the researcher.

## DISCUSSION

Physicians' decision to prescribe statins was influenced by factors operating at the microlevel and macrolevel which mapped well on the framework developed by Scoggins *et al* (2006) in a study on the factors influencing GPs prescribing of statins in primary care in England. The key microlevel factor was a set of patient's characteristics, especially the high level of TC. At the macrolevel, physicians explained that their knowledge on prescribing was grounded in their education at university as well as reading guidelines or other reference sources such as medical journals. Another macrolevel factor was the new NHIS which facilitated or limited the prescribing of statins.

The strength of this study lies in the use of a qualitative method to explore processes behind the behaviour of prescribing statins in Indonesia, an LMIC in transition to achieve UHC. The validity of the data was improved by follow-up interviews, member checking process, independent coders and discussion with several researchers to reach consensus. Although we have reached code saturation with our interviews, we only interviewed a relatively small number of physicians having relatively homogenous characteristics from one city in one province in Indonesia. Furthermore, the interviewer knew a number of the physicians personally and this may have had an impact on the willingness to share sensitive information. The interviewer had the subjective impression that those interviews were longer and contained more information than the interviews with the physicians which she met the first time. Owton and Allen-Collinson discussed the use of friendship approach for data collection in ethnographic research. Although they noted interactional challenges during interviews, for example, emotional overload and power struggle, due to a role conflict being a friend and a researcher, but overall, this approach revealed more personal and sensitive information in some encounters.[43] Nevertheless, several of our findings are comparable with other qualitative studies. Therefore, although the current study is explorative, our findings are the foundation for a future quantitative study, for example, surveys and interventions, in Indonesia. More

qualitative work on eliciting factors influencing physician's prescribing behaviour in other Asian regions and other therapeutic areas could complement our findings.

Similar to Scoggins *et al* (2006), our study found the clinical need of patients as an important factor to consider prescribing statins. However, physicians in our study described the clinical needs in more detail, mainly the level of TC and other risk factors potentially increasing the risk of CVD. Of note, the physicians did not specifically differentiate between statin prescribed for primary or secondary prevention of CVD. Interestingly, the range of cholesterol levels to start statins ranged from 200 mg/dL to 260 mg/dL and there was also a variation in taking other CVD risk factors into account. For example, one physician decided to give a statin when the TC level ≥210 mg/dL and the patient also had clinical symptoms and comorbidities while another physician only prescribed statins when the patient's TC level was ≥260 mg/dL without the presence of comorbidities. This practice was not in line with the local guideline to initiate statin therapy.[2] The patient's total CV risk calculation was only acknowledged once after probing. This variation may lead to underprescribing or overprescribing of statins. The hesitation and inconsistency in using risk assessment tools as a guidance to initiate statin therapy have also been revealed in another qualitative study performed among GPs in the UK which showed that GPs found it difficult to interpret the tool for Asian patients and therefore preferred to assess the risk intuitively.[44] The physicians in our study had some general background knowledge on statin prescribing which was by and large in line with the National Formulary. They did not mention the use of the national guideline issued by the IHA as sources to guide statin prescribing. This raises the question whether this guideline is sufficiently promoted.

Our general findings are also in agreement with a report from WHO on the drug use situation in Indonesia in 2011. It reported non-uniform sources of drug information used by physicians; they could be old text books from medical school or the internet with little use of up-to-date sources of independent drug information. The national standard treatment guideline for PHCs was not used much by physicians.[45] Even if it was used, there was no recommendation to use statins in patients with coronary heart disease as the guideline was developed when statins had not routinely been used in clinical practice.[46]

We could not disclose the role of medical representatives or pharmaceutical companies as shown in other studies[24 42] in influencing statin prescribing. Most likely the interviewees perceived this topic as too controversial within both the medical profession and broader community to be willing to discuss it.[47] Nevertheless, the WHO report suggested medical representatives from pharmaceutical companies as one of probable sources of drug information used by most physicians.[45] As the WHO report is based on a situational analysis conducted in 2011 where the NHIS was still not integrated, it is not possible to know to what extent the role of pharmaceutical companies has changed. As this question clearly created an ethical dilemma, therefore, a

collaboration with the Ministry of Health and professional bodies and councils is needed to provide a less biased answer.

Patients' demand or request for medicines is a factor influencing physicians' prescribing behaviour according to Scoggins *et al* (2006). This factor was not mentioned in our interviews although there was an indication that patient's insistence may influence the physician to meet the patient's expectation, for example, in granting a referral. The lack of involvement of the patients' voice in the physicians' decision can be caused by differences in physician–patient communication style which may be rooted in cultural differences between Southeast Asians and Westerners.[48 49] The majority of studies exploring factors influencing physicians' prescribing behaviour come from Western countries where the physician–patient communication style is often based on a partnership style with the concept of equity.[20 24 42 44 50 51] Southeast Asian physicians commonly tend to communicate in a paternalistic or one-way style, which gives the physician a more dominant role towards the patient. The culture of Southeast Asian physicians in keeping a social distance supports this communication style.[48 49]

We found that the NHIS had a considerable influence on physicians' prescribing decisions, in particular on the choice of generics. This is in line with the concept from Scoggins *et al* (2006) that prescribing decisions are influenced by factors operating at national context or macrolevel. However, Scoggins *et al* (2006) only list national and international evidence based on authoritative journals, national guidelines, national media and the development of new treatments as factors. This difference may be explained because of differences in organising healthcare and reimbursement of medication between Indonesian and England.

In contrast to Scoggins *et al*,[42] we did not identify any factors at the mesolevel such as benchmarking between different primary care practices, collaborations between primary care and specialists and regional formularies. This is understandable as strengthening the primary care sector is a recently developed strategy with the new NHIS in Indonesia. Therefore, activities which are common practice in the English primary care system at the mesolevel have not been established in Indonesia yet. This finding is an opportunity for the government to work on the mesolevel to strengthen rational prescribing behaviour.

The findings from our study indicate that physicians have a lack of awareness of specific guideline recommendations, especially on the use of LDL-c level and total CV risk calculation as decisive factors to prescribe statins. Improved access to statins facilitated by the government through the new NHIS seemed to be a good first step in fostering rational prescribing. A next step would be to include the recommendations from clinical guidelines issued by IHA into this system, especially through the National Formulary.

**Author affiliations**
¹PharmacoTherapy, -Epidemiology & -Economics, University of Groningen, Groningen, The Netherlands
²Centre for Medicines Information and Pharmaceutical Care, Faculty of Pharmacy, Universitas Surabaya, Surabaya, Indonesia
³Department of Clinical and Community Pharmacy, Faculty of Pharmacy, Universitas Surabaya, Surabaya, Indonesia
⁴Faculty of Pharmacy, Universitas Surabaya, Surabaya, Indonesia
⁵Optima Pharmacy, Surabaya, Indonesia
⁶Department of Cardiovascular Medicine, Udayana University, Denpasar, Bali, Indonesia
⁷Department of Clinical Pharmacy & Pharmacology, University Medical Center Groningen, Groningen, Indonesia

**Acknowledgements** The authors would like to thank to all physicians participated in this study.

**Contributors** SI, SP and IWW conceived and designed the study. SP carried out the interviews. SI, SP, RR, EH and KT analysed and interpreted the data. SI and SP drafted the manuscript. IWW, BW, EH and KT gave critical inputs to the intellectual content in the drafted manuscript. All authors read, revised and approved the final manuscript. SP had full access to all of the data in the study. All authors can take responsibility for the integrity of the work.

**Funding** SI received support from the Indonesia Endowment Fund for Education (Lembaga Pengelola Dana Pendidikan), the Ministry of Finance of Republic of Indonesia, in the form of scholarship for her PhD program.

**Competing interests** None declared.

**Patient and public involvement statement** There was no patient and public involvement in the design, conduct, or reporting of our research.

**Patient consent for publication** Not required.

**Ethics approval** This study was approved by the local Indonesian Medical Association. The aim of the study was known to all participants before the interview was started. A written informed consent was obtained from each participants. Participant's personal identities were kept anonymous and all transcripts were stored and maintained confidential.

**Provenance and peer review** Not commissioned; externally peer reviewed.

**Data availability statement** Data from the interviews that correspond to the results of this study, after de-identification of study participants, can be made available upon request at the beginning 9 months and ending 36 months following manuscript publication for researchers who provide a methodologically sound proposal. Proposal should be directed to the corresponding author of this study ( sylviirawati.2010@gmail.com).

**ORCID iD**
Sylvi Irawati http://orcid.org/0000-0001-6278-9017

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
