## [Reviewer comments · BMJ Open]

ARTICLE DETAILS

TITLE (PROVISIONAL)	Key factors influencing the prescribing of statins: a qualitative study among physicians working in primary health care facilities in Indonesia
AUTHORS	Irawati, Sylvi; Prayudeni, Sari; Rachmawati, Riris; Wita, I; Willfert, Bob; Hak, Eelko; Taxis, Katja

VERSION 1 – REVIEW

REVIEWER	Dr Samuel Finnikin University of Birmingham, England
REVIEW RETURNED	28-Nov-2019

GENERAL COMMENTS	Thank you for asking me to review this interesting qualitative study exploring the factors influencing the prescribing of statins in primary health care in Indonesia. This was a clear and well written manuscript which explored an important topic in a novel healthcare setting. The introduction was well written and set the scene well. However, I do think that there needs to be a clear distinction in the introduction, and throughout the manuscript, between prescribing stains for primary or secondary prevention indications. The prescribing decisions in these cases are, at least in the UK, very different as the evidence of benefit is much greater in secondary prevention. I am unclear from the report whether the clinicians spoken to were reflecting on their practice for primary or secondary prevention, or whether they were speaking of both situations. Please could this be clarified. Similarly, in the introduction, under and over prescribing is dependent on the indication (over prescribing is not really a problem for secondary prevention). The methods were clearly explained and, although I am not an expert in qualitative methods, the methods used seemed robust. My only significant concern is the selection of participants. It is stated that the sampling was purposive, but also convenience was a significant contributor to selection. The relationship between the researcher (SP) was not made clear and there was little reflection on the impact that existing relationships may have had on the data. This needs exploring further I feel. The results are well presented. There is no indication of how long the interviews lasted; and the data presented gives the impression that there was not a lot of content to the interviews. I was surprised at how few topic areas were presented and I wonder if the rather homogeneous group of clinicians (mainly young and female) resulted in data saturation being prematurely reached. I would perhaps have expected results that would map onto more of
---

	the model provided, such as professional experience, training and peers; (but that may be due to my Western cultural bias.) However, the revelation that interviewees were reluctant to discuss the impact of pharmaceutical companies is interesting and warrants more discussion. This may reveal that interviewees had more to say. The discussion was good. It was useful to consider the areas of the model not identified from the data and the overall conclusion was appropriate. Thank you for including Figure 2 to give some context of the Indonesian healthcare system.
--	--

REVIEWER	Jeremy Sussman VA Ann Arbor and University of Michigan, USA
REVIEW RETURNED	18-Jan-2020

GENERAL COMMENTS	This is a qualitative study of the determinants of statin prescribing in a small region in Indonesia. It appears to be planning data for quantitative project. In general, the authors did an excellent job with what they had. They grounded their work on a major Rand report on primary care prescribing plus a fairly strong theoretical background. The figures are generally quite interesting and show how their setting is and is not different from other studies on related topics. The structure is sound, the writing isn't perfect, but it's pretty good and will be fine with some normal editing. It's clear the authors learned a lot from doing it and this will inform their future work. The limitation of the project are fairly clear. They interviewed only 10 providers, and they pretty clearly seem to be friends and friends of friends. This is obviously exceedingly not diverse, they were even all almost the same age. It also worried me if there might be problems of participants trying to support the investigators or other attempts to please friends. Furthermore, they didn't get especially insightful responses. They found some very specific problems, including a fairly large problem of non-matching recommendations from the governmental organization that approves formularies and the main clinical practice guidelines. Otherwise, they found people struggling with issues of comorbidity etc, that you'd expect. The data isn't rigorous to be used quantitatively, but wasn't all that insightful on it's own. I do think it will do a great job of informing future surveys and interventions.
---

	I enjoyed reading it and hope the authors are able to continue their work and increase it's scope.
--	--

VERSION 1 – AUTHOR RESPONSE

Response to Reviewer 1:

Thank you for your review of our paper. We have addressed each of your points below.

1. INTRODUCTION

Reviewer's comment:

The introduction was well written and set the scene well. However, I do think that there needs to be a clear distinction in the introduction, and throughout the manuscript, between prescribing statins for primary or secondary prevention indications. The prescribing decisions in these cases are, at least in the UK, very different as the evidence of benefit is much greater in secondary prevention. I am unclear from the report whether the clinicians spoken to were reflecting on their practice for primary or secondary prevention, or whether they were speaking of both situations. Please could this be clarified. Similarly, in the introduction, under and overprescribing is dependent on the indication (overprescribing is not really a problem for secondary prevention).

Authors' response:

We agree that there is a difference in how the aim to achieve primary and secondary prevention of CVD may affect the way of prescribing statins. We clarified the differences in the Introduction, Methods, and Discussion section.

The changes made in the manuscript:

INTRODUCTION (p.1 and 2)

The use of statins has been endorsed by local and various international guidelines¹⁻⁶ ***to prevent cardiovascular disease (CVD)-related events or death in individuals without history of previous CVD (primary prevention) or with established CVD (secondary prevention)***.⁷⁻⁹ But many studies demonstrate underprescribing, and in some cases overprescribing of this drug class. Underprescribing of statins is ***especially an issue in primary prevention***, for example in the groups with moderate-to-very high risk of CVD in the European general population and in an outpatient setting in the United States.¹⁰⁻¹³

Another study in ambulatory diabetes patients ***with established CVD*** in Surabaya revealed the phenomenon of underprescribing of secondary prevention ***with 43% of eligible patients not receiving statins***.¹⁶

Nevertheless, one case study in a United Kingdom population with a $\geq 20\%$ of 10-y risk of developing CVD, ***indicating the use of statins for primary prevention of CVD***, illustrates that underprescribing will substantially reduce quality-adjusted life years (QALYs) gained over a 9 year period (2006-2014) and overprescribing will certainly increase the cost burden of prescription between £80 million and £117 million annually without certainty whether the benefit of prevented CVD events outweighs the adverse drug reaction.¹⁷

An estimation of the total cardiovascular (CV) risk score, which is necessary to guide statin prescribing for primary prevention of CVD, is not mentioned.^{29 30}

This policy is different compared to the new clinical guideline for dyslipidemic patients issued by the Indonesian Heart Association (IHA) which adapted guidelines issued by the European Society of Cardiology (ESC) and the European Atherosclerosis Society (EAS).^{2 31} **In these guidelines, statins should be prescribed according to both the level of LDL-C and the total CV risk score, especially in patients with unknown or no history of CVD.**

METHODS ('Participants', 1st paragraph, p.7)

Indonesian regulation allows physicians to conduct their practice at a maximum of three health care facilities (public or private, primary or secondary level).³⁸ The views of the physicians included in our study reflect their practice to care for patients with and without CVD with possible indications for primary and secondary prevention of CVD.

DISCUSSION (3rd paragraph, p. 13)

Similar to Scoggins et al. (2006), our study found the clinical need of patients as an important factor to consider prescribing statins. However, physicians in our study described the clinical needs in more detail, mainly the level of TC and other risk factors potentially increasing the risk of CVD. **Of note, the physicians did not specifically differentiate between statins prescribed for primary or secondary prevention of CVD.**

2. METHODS:

Reviewer's comment:

The methods were clearly explained and, although I am not an expert in qualitative methods, the methods used seemed robust. My only significant concern is the selection of participants. It is stated that the sampling was purposive, but also convenience was a significant contributor to selection. The relationship between the researcher (SP) was not made clear and there was little reflection on the impact that existing relationships may have had on the data. This needs exploring further I feel.

Author's response:

Indeed, the selection of participants is important in data collection. We added a more elaborate clarification of this sampling in the Method section of our manuscript. We further added some reflection on the consequence of this sampling method in the Discussion section.

The changes made in the manuscript:

METHODS ('Participants', page 7)

Therefore, we purposively selected physicians who worked at the primary health care facilities in the capital city who were known to a member of the research team (SP) **in a personal (friendship) or professional (acquaintance) manner**, and consented to participate in the study.

DISCUSSION (2nd paragraph, p. 13)

Although we have reached code saturation with our interviews, we only interviewed a relatively small number of physicians **having relatively homogenous characteristics** from one city in one province in Indonesia. **Furthermore, the interviewer knew a number of the physicians personally and**

*this may have had an impact on the willingness to share sensitive information. The interviewer had the subjective impression that those interviews were longer and contained more information than the interviews with the physicians which she met the first time. Owton and Allen-Collinson discussed the use of friendship approach for data collection in ethnographic research. Although they noted interactional challenges during interviews, e.g. emotional overload and power struggle, due to a role conflict being a friend and a researcher, but overall, this approach revealed more personal and sensitive information in some encounters.*⁴³

3. RESULTS:

Reviewer's comment:

The results are well presented. There is no indication of how long the interviews lasted; and the data presented gives the impression that there was not a lot of content to the interviews. I was surprised at how few topic areas were presented and I wonder if the rather homogeneous group of clinicians (mainly young and female) resulted in data saturation being prematurely reached. I would perhaps have expected results that would map onto more of the model provided, such as professional experience, training and peers; (but that may be due to my Western cultural bias.) However, the revelation that interviewees were reluctant to discuss the impact of pharmaceutical companies is interesting and warrants more discussion. This may reveal that interviewees had more to say.

Authors' response:

We have added more information to our results on the characteristics of physicians involved in the study. As you can see, the participants were well representative of the physicians that advice patients on statin adherence.

Regarding the role of pharmaceutical companies, we changed the wording of our statement because although interviewees had something to say, they were not comfortable with this matter to be recorded and reported. Therefore, we also used WHO data from the Indonesian setting to give a more broader picture on this matter.

The changes made in the manuscript:

RESULTS (1st paragraph, p. 8 and 9)

Ten physicians (nine females, one male) were willing to participate in this study. ***The mean age of physicians was 31.7 years old (ranged 24 to 56).*** Five physicians graduated between 2010 and 2015, four between 2000-2010, and one between 1980 and 1990. ***The mean duration of medical education was 6.6 years (ranged 5 to 12 years).*** ***All physicians were GPs. One was in the process of becoming a specialist in internal medicine. All physicians were also working at more than one type of health care facility, four of them practiced at three different health care facilities including at the secondary level. On average, the physicians examined around 99 patients per week (range 30 to 250), the mean number of patients with CVD risk was 26.0 patients weekly (range 5 to 100).***

DISCUSSION (2nd paragraph, p. 14)

We could not disclose the role of medical representatives or pharmaceutical companies as shown in other studies ^{24 42} in influencing statin prescribing. Most likely the interviewees perceived this topic as too controversial within both the medical profession and broader community to be willing to discuss it.⁴⁷

4. DISCUSSION

Reviewer's comment:

The discussion was good. It was useful to consider the areas of the model not identified from the data and the overall conclusion was appropriate. Thank you for including Figure 2 to give some context of the Indonesian healthcare system.

Authors' response:

We appreciate your positive comments on this section.

Response to Reviewer 2:

Reviewer's comment:

This is a qualitative study of the determinants of statin prescribing in a small region in Indonesia. It appears to be planning data for quantitative project. In general, the authors did an excellent job with what they had. The grounded their work on a major Rand report on primary care prescribing plus a fairly strong theoretical background. The figures are generally quite interesting and show how their setting is and is not different from other studies on related topics. The structure is sound, the writing isn't perfect, but it's pretty good and will be fine with some normal editing. It's clear the authors learned a lot from doing it and this will inform their future work. The limitation of the project are fairly clear. They interviewed only 10 providers, and they pretty clearly seem to be friends and friends of friends. This is obviously exceedingly not diverse, they were even all almost the same age. It also worried me if there might be problems of participants trying to support the investigators or other attempts to please friends. Furthermore, they didn't get especially insightful responses. They found some very specific problems, including a fairly large problem of non-matching recommendations from the governmental organization that approves formularies and the main clinical practice guidelines. Otherwise, they found people struggling with issues of comorbidity etc, that you'd expect. The data isn't rigorous to be used quantitatively, but wasn't all that insightful on it's own. I do think it will do a great job of informing future surveys and interventions. I enjoyed reading it and hope the authors are able to continue their work and increase it's scope.

Authors' response:

Thank you for your clear feedback. We elaborated a bit more on the potential impact of the sampling of the participants in the Discussion section. See also reviewer 1, point 2. We also added the fact that the current study is explorative and results can be used to inform future surveys and interventions.

The changes made in the manuscript:

DISCUSSION (2nd paragraph, p. 13, 14)

Although we have reached code saturation with our interviews, we only interviewed a relatively small number of physicians **having relatively homogenous characteristics** from one city in one province in Indonesia. **Furthermore, the interviewer knew a number of the physicians personally and this may have had an impact on the willingness to share sensitive information. The interviewer had the subjective impression that those interviews were longer and contained more information than the interviews with the physicians which she was met the first time. Owton and Allen-Collinson discussed the use of friendship approach for data collection in ethnographic research. Although they noted interactional challenges during interviews, e.g. emotional overload and power struggle, due to role conflict being a friend and a researcher, overall, this approach revealed more personal and sensitive information in some encounters.**⁴³

Therefore, **although the current study is explorative**, our findings are the foundation for a **future** quantitative study, **e.g. surveys and interventions**, in Indonesia.